# Ionic Liquid-Polypyrrole-Gold Composites as Enhanced Enzyme Immobilization Platforms for Hydrogen Peroxide Sensing

**DOI:** 10.3390/s19030640

**Published:** 2019-02-03

**Authors:** Meng Li, Jing Wu, Haiping Su, Yan Tu, Yazhuo Shang, Yifan He, Honglai Liu

**Affiliations:** 1Key Laboratory for Advanced Materials, School of Chemistry and Molecular Engineering, East China University of Science and Technology, Shanghai 200237, China; limeng199273@163.com (M.L.); jwubo@connect.ust.hk (J.W.); suhaiping@163.com (H.S.); tuyan@163.com (Y.T.); liuhonglai@163.com (H.L.); 2Department of Biotechnology, School of Sciences, Beijing Technology and Business University, Beijing 100048, China

**Keywords:** ionic liquids (ILs), polypyrrole (PPy), gold particles, horseradish peroxidase (HRP), enzyme immobilization, hydrogen peroxide sensor

## Abstract

In this work, three different aqueous solutions containing imidazole-based ILs with different alkyl chain lengths ([C_n_mim]Br, *n* = 2, 6, 12) were adopted as the medium for the synthesis of ionic liquid-polypyrrole (IL-PPy) composites. Herein, the ILs undertook the roles of the pyrrole solvent, the media for emulsion polymerization of PPy and PPy dopants, respectively. The electrochemical performances of the three IL-PPy composites on a glassy carbon electrode (GCE) were investigated by electrochemical experiments, which indicated that [C_12_mim]Br-PPy (C_12_-PPy) composites displayed better electrochemical performance due to their larger surface area and firmer immobilization on the GCE. Further, C_12_-PPy/GCE were decorated with Au microparticles by electrodeposition that can not only increase the conductivity, but also immobilize sufficient biomolecules on the electrode. Then, the obtained C_12_-PPy-Au/GCE with outstanding electrochemical performance was employed as a horseradish peroxidase (HRP) immobilization platform to fabricate a novel C_12_-PPy-Au-HRP/GCE biosensor for H_2_O_2_ detection. The results showed that the prepared C_12_-PPy-Au-HRP/GCE biosensor exhibited high sensitivity, fast response, and a wide detection range as well as low detection limit towards H_2_O_2_. This work not only provides an outstanding biomolecule immobilization matrix for the fabrication of highly sensitive biosensors, but also advances the understanding of the roles of ILs in improving the electrochemical performance of biosensors.

## 1. Introduction

Sensitive and selective detection of hydrogen peroxide (H_2_O_2_) is of significance in the food industry, clinical diagnosis, environmental monitoring and biological metabolism studies [1,2]. Among the various H_2_O_2_ detection techniques, electrochemical sensors have been proved to be a promising method due to their simplicity, efficiency and high sensitivity [3]. Particularly, enzyme-based electrochemical biosensors arouse considerable interest in the sensing area because of their high catalytic activity, specificity and biocompatibility [4]. Horseradish peroxidase (HRP) is the most commonly used enzyme for fabricating H_2_O_2_ biosensors [5], however, one of its drawbacks is that the direct electron transfer of HRP on a bare electrode is shielded by the protein shells surrounding the redox center of HRP, the another is its unstable bioactivity and difficult reutilization of the native enzyme [6,7]. Enzymatic immobilization on nanoparticles or nanomaterials is expected to solve these problems. Therefore, the construction of immobilized enzyme layers that can accelerate the electron transfer as well as retain their specific bioactivity is highly desired [8]. So far, a large number of nanomaterials have been employed as substrate materials to immobilize enzymes for the construction of electrochemical enzyme sensors, such as metal nanoparticles [9], carbon nanomaterials [10], graphene [11] and conducting polymers [12]. Among these substrate materials, conducting polymers have intrinsic superiority in the electrochemical field owing to their good conductivity, easy modification and biocompatibility [13]. As one of the most promising conducting polymers, polypyrrole (PPy), consisting of five membered heterocyclic rings, is widely used for constructing biosensors on account of its prominent biocompatibility, redox properties, good environmental stability, and facile immobilization of biomolecules [14]. In order to make the most of the advantages of PPy in the fabrication of biosensors, microstructured/nanostructured PPys with high surface area and enhanced electrochemical capacitance have been successfully synthesized by chemical polymerization with the aid of additives such as organic solvents, surfactants and ionic liquids [15,16,17]. It is noteworthy that Shen et al. creatively employed 1-butyl-3-methylimidazolium tetrachloroferrate (Bmim[FeCl_4_]), an ionic liquid with a shorter alkyl chain, as the oxidant to synthesize a porous polypyrrole (PPy) film by facile interfacial polymerization, and it was found that the morphology of PPys changed from dense microspherical/nanospherical agglomerated structure to porous structures with the increasing concentration of IL, and the electrochemical performance tests showed that PPy with a porous structure displayed the highest specific capacitance and a good capacitive behavior [18]. Obviously, ILs show outstanding potential in the synthesis of PPy with good electrochemical performance that can be significantly affected by the properties of ILs such as concentration, charges and hydrophobicity. However, almost all of those studies focus on ILs with shorter chains and the understanding for the mechanism of ILs promoting the electrochemical performance of PPy is insufficient by far, making further systematically studies still necessary.

Compared with ILs with shorter chains, ILs with longer chains may exhibit more novel performance in the synthesis of PPy, because the hydrophobic interaction between ILs and PPy cannot be ignored in addition to the possible existence of π–π interactions and hydrogen bonds. The ILs with longer alkyl chains are known as surfactant ILs because they possess properties of both surfactants and ILs. The unique properties of surfactant ILs endow them with multifunctionality in the preparation of nanomaterials. Surfactant ILs can not only act as the reactants and solvents, but also be used as soft templates to regulate the size and morphology of the nanomaterials because the surfactant ILs molecules can form special microstructures such as micelles, vesicles and liquid crystals depending on their concentration. A great number of nanomaterials such as nanostructured metal materials [19], nanocrystalline metal oxide [20], nanometer-sized inorganic oxides [21] and carbon nanomaterials [22] have been prepared or functioned by using surfactant ILs. Unfortunately, it is still an information gap in the utilization of surfactant ILs in polymerization reactions of conducting polymers, and little is known about the effect of the hydrophobic chain length of surfactant ILs on the electrochemical performance of conducting polymers, which is regarded as a significant influencing factor of the resulting material properties. Besides, based on the intrinsic high conductivity and wide potential window of ILs [23], the combination of ILs and PPy is expected to improve the sensitivity of electrochemical sensors by their synergistic effect. 

It is well known that noble metal nanomaterials can be a preferable cooperator of conducting polymers for fabricating high sensitive biosensors owing to its higher conductivity, better suitability and larger specific surface area [24,25,26]. In particular, gold nanoparticles (AuNPs) are a well-known bio-nanomaterial used for biomolecule immobilization, because they can act as biomolecule carriers and electron transfer promoters [27]. Furthermore, AuNPs also can conjugate with biomaterials strongly and thus further increase the surface area and electric conductivity of the composite [28,29]. As a matter of course, AuNPs are preferred by researchers in the course of developing biosensors based on conducting polymer and noble metal hybrid materials. For instance, Rong et al. synthesized a polypyrrole (PPy) hydrogel/Au nanocomposite as an enhanced electrochemical biosensor for ultralow limit of carcinoembryonic antigen (CEA) detection [30]. Pineda et al. developed an electrochemical sensor composed of microtubular structured polypyrrole (PPy) films decorated with gold nanoparticles by an electrochemical synthesis method for the sensitive determination of hydroxylamine, nitrite and their mixtures [31]. Undoubtedly, the combination of PPy and Au particles is a feasible and efficient way to improve the sensitivity of biosensors. 

Inspired by the previous research work, we propose in this work a novel enzyme-based electrochemical biosensor that takes advantage of IL-PPy-Au composites to immobilize HRP on a GCE. The prepared C_12_-PPy-Au-HRP/GCE biosensor exhibited outstanding electrochemical performance for H_2_O_2_ detection. Herein, the IL-PPy composites were simply prepared by polymerizing pyrrole in aqueous solutions containing the ILs 1-ethyl-3-methylimidazolium bromide ([C_2_mim]Br), 1-hexyl-3-methylimidazolium bromide ([C_6_mim]Br), and 1-dodecyl-3-methyl-imidazolium bromide ([C_12_mim]Br), respectively. The effects of the hydrophobic properties of the ILs on the electrochemical performance of the prepared IL-PPy composites were investigated in detail and the mechanism of ILs in improving the sensitivity of biosensor was highlighted. The predominant [C_12_mim]Br solution was singled out as the best medium for polymerizing pyrrole and the desired conductive film composed of [C_12_mim]-PPy (C_12_-PPy) composite was formed on the GCE. Subsequently, Au microparticles were integrated onto C_12_-PPy/GCE by electrodeposition that can not only increase the electrical conductivity, but also firmly immobilize sufficient HRP molecules on the electrode. Combining the superior features of C_12_-PPy composite and Au particles, an enhanced HRP immobilization platform was fabricated successfully. Under the optimal operating potential, pH and amount of HRP immobilized on the electrode, the C_12_-PPy-Au-HRP/GCE biosensor exhibits much wider detection range and higher sensitivity for H_2_O_2_ detection. 

## 2. Materials and Methods

### 2.1. Material

Pyrrole, ammonium persulfate, horseradish peroxidase (HRP, EC1.11.1.17, 190 unit·mg^−1^) were purchased from Tokyo Chemical Industry Co., Ltd. (Tokyo, Japan). 1-Ethyl-3-methylimidazolium bromide ([C_2_mim]Br, 99%), 1-hexyl-3-methylimidazolium bromide ([C_6_mim]Br, 99%), and 1-dodecyl-3-methylimidazolium bromide ([C_12_mim]Br, 99%) were provided by Monils Chem. Eng. Sci. & Tech. (Shanghai) Co. Ltd. (Shanghai, China). K_4_Fe[(CN)_6_], K_3_Fe[(CN)_6_], Na_2_HPO_4_, NaH_2_PO_4_, KCl, citric acid and 30% hydrogen peroxide (H_2_O_2_) solution were obtained from Shanghai Lingfeng Chemical Reagent Co., Ltd. (Shanghai, China). Hydrogen tetrachloroaurate hydrate (HAuCl_4_, Au 28.6%) was provided by Shanghai Macklin Biochemical Co. Ltd. (Shanghai, China). Ascorbic acid (AA), D-(+)-glucose and uric acid (UA) were obtained from Aladdin Industrial Co., Ltd. (Shanghai, China). Sodium deoxycholate (NaDC, 98%) and aspartic acid (Asp, 98%) were purchased from Shanghai J&K Scientific (Shanghai, China). Ultrapure water with a resistivity of 18.2 MΩ·cm from a Simplicity water purification system (Millipore, Boston, MA, USA) was used for preparing buffer solutions with different pHs. All other chemicals were of analytical reagent grade and used without further purification.

### 2.2. Instruments

SEM images and EDS were obtained from a Nova Nano SEM 450 system (FEI, Hillsborough, OR, USA) equipped with the TEAMEDS energy dispersive spectrometer. The studied samples were casted on the surface of glassy carbon electrodes customized for electron microscopy (Chuxi Instruments Co., Shanghai, China) to form a thin film layer. The dehydrated specimens were sprayed with platinum before examination. FTIR spectra were measured by a Nicolet 6700 FTIR spectrometer (Thermo Fisher, Waltham, MA, USA). The samples were prepared by the KBr disc or film technique. The Zeta potential of different samples were determined by dynamic light scattering (DLS) (Zem4228, Malvern Instruments, Malvern, UK). The specific surface areas of different samples were calculated by the Brunauer–Emmett–Teller (BET) method (ASAP-2020, Micromeritics, Atlanta, GA, USA). All electrochemical measurements were carried out on a CHI660E electrochemical workstation (Chenhua Instruments Co., Shanghai, China). The three-electrode system was composed of a working electrode (GCE, 3.0 mm in diameter), a platinum wire counter electrode and a silver chloride reference electrode.

### 2.3. Synthesis of IL-PPy Composites

The IL-PPy composites were prepared by the following steps: firstly, pyrrole was dissolved in aqueous solutions (1 mL, 1 M) of [C_2_mim]Br, [C_6_mim]Br or [C_12_mim]Br, respectively, then the above solutions were sonicated uniformly to obtain 1 mM pyrrole solutions. Then ammonium persulfate solution (250 μL, 1 M) that acted as initiator was quickly added to the above freshly prepared mixture solutions, which were then placed in the dark for 1 h to form the black IL-PPy composites as pyrrole autopolymerizes to form a large amount of oligomers under light. The resulting IL-PPy composites were subsequently centrifuged, washed with ultrapure water and ethanol to remove by-products formed during polymerization and excess ILs. Finally, the obtained IL-PPy composites were dried under vacuum at 60 °C. The same procedures were carried out in the control experiment using water as the pyrrole polymerization medium. For simplicity, the obtained samples in water and [C_n_mim]Br (*n* = 2, 6, 12) aqueous solutions are abbreviated as PPy, C_2_-PPy, C_6_-PPy and C_12_-PPy, respectively, in the following sections.

### 2.4. Preparation of Electrochemical Enzyme Sensor

The GCE was polished repeatedly using alumina powder and then thoroughly cleaned before use. After that, 10 μL freshly prepared mixed solution consisting of IL, pyrrole and ammonium persulfate were dropped on the GCE and then dried in the dark for 20 min for formation of a thin IL-PPy film. Subsequently, the modified electrode was flushed with ultrapure water to remove excess ions and any oligomers. An IL-PPy composite-modified electrode was obtained whereby a shiny black film can be observed on the surface of GCE. 

C_12_-PPy composite was singled out to be further decorated with Au. Au particles were integrated on the prepared C_12_-PPy/GCE by electrochemical deposition according to the as-reported method with some modification [30]. Briefly, the as-prepared C_12_-PPy/GCE was processed by cyclic voltammetry scanning from 0.2 V to −1.0 V in 1 mM HAuCl_4_ solution containing 0.1 M KCl at a scan rate of 50 mVs^−1^ for 30 segments. After that, a golden surface appeared on the modified electrode that indicated the formation of Au. Following that, HRP molecules was immobilized on the C_12_-PPy-Au/GCE by directly dropping 10 μL of aqueous HRP solution on the modified electrode which was kept overnight at 4^o^C. In this way, HRP molecules were absorbed on the modified electrode by electrostatic interaction between HRP and Au [32]. After each modification, the modified electrode was thoroughly rinsed with ultrapure water to remove any unattached species. Finally, the desired enzyme-based electrochemical sensor of C_12_-PPy-Au-HRP/GCE was obtained and stored at 4 °C for use.

### 2.5. Electrochemical Measurements

All of the electrochemical measurements were carried out at room temperature and the electrolyte solutions were sufficiently deaerated with high purity nitrogen. The electrodes modified by different materials were measured by cyclic voltammetric measurements (CV) scanning from −0.5 V to 0.2 V (vs.Ag/AgCl) in 0.01 M PBS (pH 7.0) containing 5.0 mM [Fe(CN)_6_]^4−^/^3−^ and 0.1 M KCl with 50 mV·s^−1^. Differential pulse voltammetry (DPV) was also adopted to monitor the formation details of the enzyme-based electrochemical sensors; the measurements were conducted from −0.5 V to 0.1 V (vs.Ag/AgCl) with a pulse amplitude of 50 mV and a pulse width of 50 ms. 

Electrochemical impedance spectroscopy (EIS) curves were recorded within the frequency range of 0.01 Hz to 1000 KHz at the formal potential in 0.01 M PBS (pH 7.0) containing 5.0 mM [Fe(CN)_6_]^4−^/^3−^ and 0.1 M KCl. The amplitude of the alternate voltage was 5 mV. The EIS were plotted in the form of Nyquist plots.

Amperometric measurements were employed to quantitatively detect H_2_O_2_ by consequently adding H_2_O_2_ solution with a certain concentration into phosphate buffer solution (0.1 M) at −0.3 V (vs.Ag/AgCl), a magnetic Teflon stirrer provided the convective transport during the measurement. 

## 3. Results and Discussion

### 3.1. Characterization of IL-PPy Composites

The IL-PPy composites were prepared and the obtained samples were characterized by FTIR. Figure 1 depicts the FTIR spectra of different samples prepared in water, [C_2_mim]Br, [C_6_mim]Br and [C_12_mim]Br aqueous solutions, respectively (curves a-d). It can be seen that the four samples all show obvious infrared characteristic PPy peaks located at 1537 cm^−1^, 1450 cm^−1^ and 1160 cm^−1^ corresponding to the C-C, C=C and C-N stretching vibrations of the pyrrole ring, respectively [33]. The bands at 906 cm^−1^ and 1041 cm^−1^ are assigned to the C-C out-of-plane vibration and C-H in-plane vibration of PPy, respectively [34]. Compared with curve a (the sample prepared in water), no significant changes can be observed in curve b (the sample prepared in [C_2_mim]Br solution), which indicates that the components of the two samples are almost the same. However, for the samples prepared in [C_6_mim]Br (curve c) and [C_12_mim]Br (curve d) solutions, the infrared absorption peaks of the methyl imidazole cation at 3100 cm^−1^, 1636 cm^−1^, 1445 cm^−1^ appear, as shown in curve c and curve d. The obvious absorption peaks at 853 cm^−1^, 752 cm^−1^ and 651 cm^−1^ can be attributed to the C-H out of plane bending vibrations of the imidazole ring [35], which reconfirms the existence of a methyl imidazole ring in the samples obtained in [C_6_mim]Br and [C_12_mim]Br solution, respectively. Upon further observation can be seen that the characteristic peaks of PPy at 1537, 1450, 1160, and 1041 cm^−1^ are all upshifted about 10–20 cm^−1^ to higher wavenumbers in curve c and curve d. This phenomenon should result from the π–π interaction and hydrogen bond between the imidazole rings of the ILs and the PPy backbone [36]. The above results indicate that ILs with longer alkyl chains ([C_6_mim]Br and [C_12_mim]Br) can take part in the polymerization of pyrrole and further form the hybrid IL-PPy composites by hydrophobic interaction, π–π interaction and hydrogen bonds between pyrrole and the ILs. Whereas only the pure PPy can be detected for the sample prepared in the solution containing an IL with a short chain ([C_2_mim]Br). Apparently, the hydrophobic interaction between pyrrole and ILs prevails. 

The surface charges carried by the particles of different samples were measured by Zeta potential as shown in Appendix A. As can be seen, the species prepared in water and ILs are all positively charged. Compared with PPy prepared in water, the introduction of ILs increases the surface positive charges of the species and the surface positive charges increase with the chain length of the ILs. The significant increase of surficial positive charges can be attributed to the [C_6_mim]^+^ or [C_12_mim]^+^ covering the surface of PPy by hydrophobic interactions, which implies the formation of C_6_-PPy and C_12_-PPy composites. This agrees well with the results obtained from IR spectra.

The prepared species in different media were used for modifying GCE customized for electron microscopy, and the dehydrated species on the electrodes were characterized by SEM (Figure 2). As can be seen in Figure 2A, PPy obtained in water forms microspheres with a diameter of about 1 μm, which are sparsely distributed on the electrode and stick to each other. Compared with the PPy spheres polymerized in water, the size of the species formed in [C_2_mim]Br solution decreases significantly and these spheres tend to aggregate together. Meanwhile, the number of formed spheres is limited as shown in Figure 2B, which may result from short-chain ILs ([C_2_mim]Br) being incompetent to immobilize PPy on the GCE firmly due to the weak hydrophobic interactions between the ILs and the hydrophobic surface of the GCE, thus a limited quantity of PPy spheres remains on the GCE during the modification procedure. However, as can be observed in the C_6_-PPy composite, there are smaller and denser spheres dispersed on the electrode compared with that obtained in [C_2_mim]Br solution. Upon further observation it can be seen that some slightly rugged films composed of numerous C_6_-PPy spheres with diameter about 100-200 nm are formed on the electrode (Figure 2C). As for the species formed in [C_12_mim]Br solution, the electrode is covered by plenty of folded films that are comprised of a large number of several dozen-nanometer C_12_-PPy spheres connected with each other (Figure 2D).

Obviously, the length of alkyl chain of the ILs affects the properties of the resulting composites. According to the above results, it is undeniable that the introduction of ILs with longer chains can not only significantly decrease the particle size of the IL-PPy composites, but also efficiently increase the amount of folded films on the electrode. All of these facts favor the enlargement of the charge transfer area on the electrode, which is beneficial for improving the conductivity of the biosensor and immobilizing more biomolecules. It is well known that the longer the hydrophobic chain length of a surfactant, the more micelles with smaller size are formed in surfactant systems [37]. Therefore, the micelles decrease in size but increase in number with the increase of chain length of ILs in the solution when the IL concentration keeps constant, which directly determines the size and the number of the IL-PPy composites particles. The existence of micelles not only improves the solubility of the oily pyrrole but also provides microreactors for the polymerization of pyrrole, which provide favorable conditions for the formation of uniform, dense and nanoscaled IL-PPy composite spheres. Moreover, stronger hydrophobic interactions, π–π interactions and hydrogen bonds between ILs and PPy lead to a firm linkage between imidazolium cations ([C_6_mim]^+^ or [C_12_mim]^+^) and PPy, which contribute to the formation of the C_6_-PPy film and C_12_-PPy film on the electrode. Based on the above discussion, the PPy-IL composite with longer hydrophobic chain, namely C_12_-PPy composite, should have the largest specific surface area. The fact is confirmed by the BET results of different samples (Appendix A), the BET surface area of samples ranked the order of C_12_-PPy > C_6_-PPy > C_2_-PPy > PPy, and the C_12_-PPy composite shows the largest BET surface area of 46.7 m^2^·g^−1^. Thus, C_12_-PPy composite is expected to exhibit a preferable electrochemical performance.

In order to investigate the effect of the alkyl chain length of ILs on the electrochemical performance of IL-PPy films, some electrochemical measurements were carried out on GCEs modified by IL-PPy (IL-PPy/GCE). Figure 3 shows the cyclic voltammograms (CVs) of different electrodes in [Fe(CN)_6_]^3−/4−^ solution. Obviously, both the cathodic and the anodic peak currents of the modified GCEs increase to a different degree compared with that of bare GCE. As we know, PPy is particularly gifted at conduction, however, the peak current of GCE only increased slightly after being modified by PPy synthesized in water (curve b). The unexpected result may be attributed to the weak adhesion between simple PPy film and the electrode, which makes it difficult to firmly immobilize the PPy film on the glossy surface of the GCE. The introduction of ILs improves the peak current of the electrode to a certain degree (curve c-e), which should be due to the tighter immobilization of the IL-PPy composites on the GCE by hydrophobic interactions between the IL-PPy and the GCE. Certainly, the co-effect of the excellent conductivity of PPy and the better electrochemical activity of ILs also promotes the conductivity of the IL-PPy composites. According to the increase of the peak current, the conductivity of IL-PPy composites can be ranked in the following order: C_2_-PPy < C_6_-PPy < C_12_-PPy, which coincides with the results of BET surface area for the different materials (presented in Appendix A). Reasonably, C_12_-PPy composites show the highest conductivity among the studied IL-PPy composites due to their larger surface area and firmer immobilization on the GCE.

Differential pulse voltammetry (DPV) is a sensitive and convenient tool to monitor the fabrication process and the performance of biosensors. The effect of the ILs on the conductivity of the IL-PPy composites was investigated by scanning DPV of the GCEs modified by different IL-PPy composites in [Fe(CN)_6_]^3−/4−^ solutions. Figure 4 shows the current responses of the GCEs before and after modification, in which the variations of current are more obvious than that of the corresponding samples in Figure 3. The results reconfirm that the combination of ILs and PPy improve the current of the electrode and the IL-PPy/GCE currents increase with the alkyl chain length of the ILs. Doubtlessly, the alkyl chain length of the ILs is the key factor affecting the conductivity of IL-PPy composites on the GCE. Scheme 1 illustrates the effect of alkyl chain length of ILs on the properties of the IL-PPy conducting films formed on GCE, in which ILs mainly play three roles. Firstly, IL solutions provide favorable reaction media for pyrrole polymerization. Especially, [C_6_mim]Br and [C_12_mim]Br can efficiently emulsify the oil-soluble pyrrole by encapsulating pyrrole in the hydrophobic core of the IL micelles, and the obtained emulsions provide beneficial conditions for the emulsion polymerization of pyrrole, which is proved to be an efficient method for synthesizing PPy nanomaterials with high specific surface area [38]. Secondly, long-chain ILs can act as dopants of PPy due to π–π interactions and hydrogen bonds as well as hydrophobic interactions between the ILs and PPy. The proportion of ILs in the IL-PPy composites increases with the increase of chain length of ILs due to the stronger hydrophobic interactions between the PPy and ILs with longer alkyl chains. Thirdly, the presence of ILs can immobilize IL-PPy composites on the GCE to some degree, and the longer chain length of the ILs, the tighter the immobilization of the IL-PPy composites on the GCEs, which should be attributed to the stronger hydrophobic interactions between the IL-PPy composites and the GCEs. Therefore, the C_12_-PPy composites with larger surface area and stronger immobilization on the GCE exhibit preferable conductivity. Furthermore, the larger surface area of the C_12_-PPy composites can not only greatly facilitate electron transfer based on PPy’s special long π-conjugated backbone and the imidazole cations of the ILs, but also provide more sites for immobilization of HRP molecules.

As we all know the anionic surface active agent NaDC is usually used as gelatinizing agent for immobilizing biomolecules owing to its special rigid structure and good biocompatibility [39]. In the present study, NaDC was selected as the substitute of the ILs to prepare the NaDC-PPy/GCE and the corresponding conductivity was further studied. The results show that the formed species carry negative charges (Appendix A), which is caused by a large number of negatively charged DC anions (DC^−^) attached on the PPy particles by electrostatic interaction. This can be regarded as indicative of the formation of NaDC-PPy composite. However, the introduction of NaDC in PPy reduces the conductivity significantly, as shown in Appendix A. Reasonably, the attached negatively charged DC^−^ may block the electron transmission channels on the π-conjugated backbone of PPy and thus decrease its conductivity [40]. Therefore, charges carried by dopants affect the conductivity of PPy samples significantly. It can be deduced that the negatively charged dopants can weaken the conductivity of PPy significantly, and on the contrary, positively charged dopants can enhance the PPy conductivity due to the fact the cations assist the PPy to capture more electrons from the electrolyte solution. In the present study, the positively charged imidazolium cations benefit the capture of more electrons from the electrolyte solution by PPy and thus increase the conductivity of PPy.

### 3.2. Electrochemical Performance of C_12_-PPy-Au-HRP/GCE

Generally speaking, the existence of AuNPs on biosensors can provide lots of electrostatic binding sites for immobilizing sufficient HRP molecules and thus improve the sensitivity of biosensors [32]. Based on the outstanding electrochemical performance of C_12_-PPy composite, C_12_-PPy/GCE was singled out for further electrodepositing Au. During the process of electrodeposition of Au particles, the distribution density of Au particles on the surface of C_12_-PPy/GCE depends on the scanning cycles of cyclic voltammetry when the concentration of HAuCl_4_ solution (1 mM) and scan rate (50 mVs^−1^) are both kept constant. Generally, the distribution density of Au particles increases with the increased number of scanning cycles. However, the current response of C_12_-PPy-Au/GCE increases firstly then tends to decrease with the increase of scanning cycles as shown in Appendix A. The reason for the decrease of the current response may be attributed to the formation of thicker Au layers on the C_12_-PPy film caused by excessive scans, which in turn hinders the electron transfer on the π-conjugated backbone of PPy. It can be seen that scanning cycles of 30 segments (15 scanning cycles) lead to a larger current response (Appendix A). Thus, the C_12_-PPy-Au/GCE was prepared by scanning C_12_-PPy/GCE for 30 segments in 1 mM HAuCl_4_ solution at a scan rate of 50 mVs^−1^. 

The obtained C_12_-PPy-Au composite was characterized by SEM and EDS (Figure 5). It is clearly shown that the original smooth folded films of C_12_-PPy composites on GCE (Figure 2D) tend to become rougher and a number of aggregated particles appear on the rougher folded films, which should be ascribed to the formation of Au microparticles on the surface of C_12_-PPy film due to the aggregation of numerous AuNPs. Additionally, three major elements namely C, N and Au are shown in the EDS images of the modified GCE (Figure 5C), which suggest that the formation of C_12_-PPy-Au composite on the GCE.

The DPV current responses of GCE with stepwise modification are presented in Figure 6. It can be seen that the DPV response of C_12_-PPy-Au/GCE increases compared with that of C_12_-PPy/GCE, which indicates that the participation of Au particles increases the conductivity by enlarging the charge transfer area of the electrode. Sequentially, the DPV response decreases sharply after further immobilization of HRP due to the fact the enzyme layers on the electrode can retard the electron transfer to a great degree. Electrochemical impedance spectroscopy (EIS) was also employed to measure the interfacial properties of electrode with stepwise modification (Figure 7). The typical Nyquist plots of EIS consist of a linear part and a semicircle part, where the linear part in the lower frequency portion represents the diffusional limited electron-transfer process, and the semicircular part in the higher frequency portion is related to electron-transfer-limited processes and the electron transfer resistance (R_et_) can be determined by the semicircle diameter. As Figure 7 displays the Nyquist plots of different modified GCEs are distinctly different. Compared with the R_et_ of a bare GCE, the R_et_ of C_12_-PPy-Au/GCE significantly decreases because the corresponding semicircle diameter (curve b) is much smaller than that of bare GCE (curve a), which indicates that C_12_-PPy-Au composites possess outstanding conductivity that can efficiently improve the electron transfer on the electrode. Besides, the Nyquist plots of C_12_-PPy-Au/GCE mainly consist of a straight line at lower frequency with a slope near 45 degrees, accompanied by a tiny semicircle at higher frequency, which characterizes that C_12_-PPy-Au/GCE shows higher reversibility and the corresponding electrochemical process is dominated by a diffusional limited electron-transfer process [41]. As for C_12_-PPy-Au-HRP/GCE, the semicircle diameter of the Nyquist plot increases significantly (curve c), which is indicative of an electron transfer hindered by nonconductive HRP molecules. Consequently, the reversibility of the electrode decreases and the electrochemical process is driven by electron-transfer-limited and diffusional limited electron-transfer hybrid processes.

Further, the responsibility of C_12_-PPy-Au-HRP/GCE to H_2_O_2_ was explored by cyclic voltammetry. Figure 8A shows the CV curves of different electrodes obtained in PBS containing 100 μM H_2_O_2_. It can be seen that the redox peak cannot be detected for the bare GCE (curve a). However, GCE modified by HRP shows a weak response current to H_2_O_2_ because HRP alone is difficult to immobilize on a bare GCE (curve b). Nevertheless, C_12_-PPy-Au-HRP/GCE shows a remarkable cathodic peak current increase in H_2_O_2_ solution (curve c), which indicates that HRP immobilized on C_12_-PPy-Au composite can efficiently catalyze the reduction of H_2_O_2_. It is worth mentioning that C_12_-PPy-Au-HRP/GCE shows a weak cathodic peak at about −0.3 V in the PBS without H_2_O_2_, which represents the reduction process of Fe(III) to Fe(II) occurring on the heme group of HRP, as reported [27]. The fact HRP molecules are immobilized on the C_12_-PPy-Au composite matrix is thus reconfirmed. 

Figure 8B displays the CV response of C_12_-PPy-Au-HRP/GCE for H_2_O_2_ concentrations. Obviously, the cathodic peak currents increase gradually with the increase of the H_2_O_2_ concentration from 0 to 500 μM, which corresponds to the reduction process of H_2_O_2_. Thereby, C_12_-PPy-Au-HRP/GCE shows not only good catalytic activity toward H_2_O_2_, but also potential applicability for the quantitative detection of H_2_O_2_.

### 3.3. Optimization of Detection Conditions

The operating potential, pH value and the amount of HRP immobilized on the electrode are influential factors of the performance of enzyme-based biosensors. Therefore, the detection conditions were further optimized to obtain a C_12_-PPy-Au-HRP/GCE sensor with excellent performance. The optimal operating potential corresponding to the maximum response current to H_2_O_2_ is optimized firstly. The potential optimization range is assigned from −0.1 V to −0.5 V according to the obtained response current in the CV curves (Figure 8B). Chronoamperometry is adopted to measure the response current of the PPy-Au-HRP/GCE biosensors in PBS containing 100 μM H_2_O_2_ at different constant potentials. The optimization result is shown in Figure 9A, in which the response currents of the biosensor increase with the increase of the potential in the range from −0.1 V to −0.3 V. However, when the potential surpasses −0.3 V, the current baseline tends to become unstable and quivering, which may result from the decrease of stability of HRP on the electrode caused by the increase of the potential. Therefore, the optimal operating potential is assigned to −0.3 V at which the subsequent optimization was conducted. 

Figure 9B depicts the result of pH optimization. It can be seen that the response current of C_12_-PPy-Au-HRP/GCE varies with pH values, and the response currents increase with the increase of pH between 4.0 to 7.0, but decrease suddenly while the pH further increases to 8.0. The result suggests that HRP has higher activity in neutral pH solution, which is consistent with results obtained by other studies on HRP-based electrochemical sensors [7,42]. Thus, the preferable pH is selected as 7.0.

Under the optimal potential and pH conditions mentioned above, the amount of HRP immobilized on the electrode was further optimized by immersing overnight the C_12_-PPy-Au/GCE into HRP solutions with concentrations from 1.0 to 9.0 mg·mL^−1^. Figure 9C provides the variation of current response of C_12_-PPy-Au/GCE with the concentration of HRP solution. The results indicate that the current responses of C_12_-PPy-Au/GCE increase significantly when the HRP concentrations increase from 1.0 to 4.0 mg·mL^−1^. With further increase of the HRP concentrations, the current responses of C_12_-PPy-Au/GCE increase slightly until the HRP concentration reaches 8.0 mg·mL^−1^. However, the current response of C_12_-PPy-Au/GCE remains almost constant once the HRP concentration is over 8 mg·mL^−1^. Thus the 8.0 mg·mL^−1^ was the optimal concentration in the present study. It was worth mentioning that the difference of current response is slight when the HRP concentration is in the range of 4.0–8.0 mg·mL^−1^, and thus the corresponding data for 5.0, 6.0 and 7.0 mg·mL^−1^ HRP solutions have not been provided in Figure 9C. 

### 3.4. Catalytic Performances of the C_12_-PPy-Au-HRP/GCE

Under the optimum conditions, a typical current–time plot of a C_12_-PPy-Au-HRP/GCE biosensor upon the successive addition of H_2_O_2_ is shown in Figure 10A. As H_2_O_2_ of increased concentration (2–60 μM) is added into the stirring buffer solution, a stepwise increase of the current response is observed. Actually, a weaker current can be detected when the concentration of H_2_O_2_ is as low as 0.25 μM, but the current baseline is unstable. The current increases stably with the addition of aliquots of H_2_O_2_ (the inset in Figure 10A) until the concentration of H_2_O_2_ reaches 2 μM. In particular, the response currents increase sharply to a stable value within 1 s by each addition of H_2_O_2_. Such a fast response should be attributed to the cooperation of C_12_-PPy composite and Au particles on GCE. On the one hand, C_12_-PPy composites greatly increase the surface area of the electrode and facilitate the electron transfer between electrode and H_2_O_2_ due to their excellent conductivity. On the other hand, Au particles are favorable to the orientation of the HRP molecules on the electrode in the bioelectro-catalysis process, which make electron transfers occur easily [43,44]. 

The calibration curve shown in Figure 10B indicates that the relationship of the currents and the H_2_O_2_ concentrations is fitted a linear equation of y = 0.0675x + 0.413 (R^2^ = 0.998), and the linear range is from 2 μM to 420 μM. The H_2_O_2_ biosensor based on C_12_-PPy-Au-HRP/GCE exhibits a sensitivity of 540 μA·mM·cm^−2^ and a detection limit of 0.25 μM (S/N = 3). Compared with other HRP biosensors, the proposed biosensor C_12_-PPy-Au-HRP/GCE exhibits better performance for the detection of H_2_O_2_ in terms of a wider linear range and lower limit of detection (Table 1), suggesting the proposed biosensor compares advantageously with respect to other biosensor designs.

Inspiringly, the prepared C_12_-PPy-Au-HRP/GCE biosensor not only exhibits a sensitive current response to H_2_O_2_ detection, but also a high affinity between HRP and H_2_O_2_. The apparent Michaelis-Menten constant ($K_{m}^{app}$) for H_2_O_2_ that gives an indication of the enzyme-substrate kinetics can be obtained from the electrochemical version of Lineweaver-Burk equation as shown in Equation (1) [49]:(1)$$1/I_{ss}=1/I_{max}+K_{m}^{app}/I_{max}C$$
where $I_{ss}$ is the steady-state current under a certain concentration of H_2_O_2_, $I_{max}$ is the maximum current measured under saturated substrate solution and C is the bulk concentration of the substrate, which are applicable for calculating $K_{m}^{app}$ of the biosensor. Consequently, the calculated $K_{m}^{app}$ of HRP in C_12_-PPy-Au-HRP/GCE for H_2_O_2_ is 0.291 mM, this value is lower than the 23.15 mM for HRP entrapped in clay–chitosan-gold nanoparticle nanocomposite modified GCE [50], 11.94 mM for HRP entrapped in graphene and dsDNA composite on the surface of a carbon ionic liquid electrode (CILE) [51] and 2.79 mM for HRP immobilized onto poly(aniline-co-*N*-methylthionine) film [52]. As we know a smaller $K_{m}^{app}$ indicates higher affinity between HRP and H_2_O_2_, the HRP in C_12_-PPy-Au-HRP/GCE biosensor exhibits a high affinity towards H_2_O_2_ due to C_12_-PPy-Au composite provide a sufficient microenvironment for enzyme-substrate interaction.

### 3.5. Selectivity and Stability C_12_-PPy-Au-HRP/GCE Biosensor

Selectivity and stability are two important factors of the performance of enzyme-based sensors. Firstly, the selectivity of C_12_-PPy-Au-HRP/GCE biosensor was investigated by sequentially adding several interferent molecules, including 1 mM ascorbic acid (AA), 1 mM glucose, 1mM uric acid (UA) and 1 mM aspartic acid (Asp) into the PBS containing 100 μM H_2_O_2_. As shown in Figure 11A the C_12_-PPy-Au-HRP/GCE biosensor exhibits negligible signals towards interferent molecules at concentrations of 10 times higher than that of H_2_O_2_, which demonstrates the proposed biosensor has excellent selectivity towards H_2_O_2_. Additionally, the stability of C_12_-PPy-Au-HRP/GCE biosensor was investigated by constantly measuring the amperometric response in PBS containing 100 μM H_2_O_2_ for 10 days (Figure 11B). The biosensor was stored in PBS (pH 7.0) at 4 ^o^C when not used. After 10 days of storage, the current response toward H_2_O_2_ of the biosensor still remained at 77.6% of its initial value, which is indicative of the firm immobilization of HRP on C_12_-PPy-Au composite matrix that provides a biocompatible microenvironment for preventing enzyme inactivation.

## 4. Conclusions

In summary, a novel H_2_O_2_ biosensor was easily fabricated by immobilizing HRP on a IL-PPy-Au composite matrix. Herein, the imidazole-based ILs play important roles in improving the conductivity of the IL-PPy composites, which undertake the roles of pyrrole solvent, medium for the emulsion polymerization of PPy and PPy dopants, respectively. The results show that the introduction of ILs with longer chains can not only significantly decrease the particle size of the IL-PPy composites but also efficiently increase the amount of folded conducting films on the electrode, which can greatly enlarge the charge transfer area on the electrode and further improve the conductivity of the biosensor. Reasonably, GCE modified by C_12_-PPy composite displays superior conductivity due to its larger surface area and firmer immobilization, on which Au microparticles are further electrodeposited for promoting electron transfer and immobilizing HRP. The prepared C_12_-PPy-Au-HRP/GCE biosensor exhibits a higher sensitivity, faster response, and a wider detection range as well as a lower detection limit towards H_2_O_2_. Obviously, the combination of [C_12_mim]Br, PPy and Au microparticles provides an enhanced platform for enzyme immobilization that is indeed an efficient method to construct high sensitivity biosensors. This work not only provides an outstanding biomolecules immobilization matrix for the fabrication of biosensors with higher sensitivity, but also advances our understanding of the role of ILs in improving the electrochemical performance of biosensors, which is of significance for design and optimization of biosensors based on ILs.

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
