# Peer review of "Ionic Liquid-Polypyrrole-Gold Composites as Enhanced Enzyme Immobilization Platforms for Hydrogen Peroxide Sensing"

_sensors, 2019, doi:10.3390/s19030640_

Reviewer 1 Report

The paper “Ionic Liquids-Polypyrrole-Gold Nanocomposite as an Enhanced Enzyme Immobilization Platform for Hydrogen Peroxide Sensing” studied the synthesis of liquids-polypyrrole-gold composites and their applications of H2O2 sensing. It is systematically studied and the paper is well written. It can be accepted after the authors address the flowing comments.

1. In the Synthesis of ILs-PPy composite, the addition of ammonium persulfate solution should be explained, and the reason why “placed in the dark” should also be explained.

2. SEM is not enough to support the large surface area of C12-PPy, BET measurement is required.

3. What is the actual amount of Au on the C12-PPy/GCE? Is it can be controlled? Similarly, what is the actual amount of HRP on the C12-PPy/GCE?

4. In Figure 5A and B, it can be seen that a lot of aggregated particles on the surface. Are they Au? If they are Au, the claim of AuNPs is not suitable, as the particles are in micro-size.

5. In Figure 9C, the concentration of HRP is 1, 2, 3, 4, 8, and 9. Why there is a big gap between 4 and 8? The optimal concentration of HRP solution of 8 may be incorrect.  In my opinion, authors should add more concentrations of HRP between 4 and 8 to get an optimal.

6. There are some typos in the manuscript. eg. line 164, “absorb” should be absorbed; line 238, “know” should be “known”.

Author Response

Response to Reviewer 1 Comments

Comments and Suggestions for Authors

The paper “Ionic Liquids-Polypyrrole-Gold Nanocomposite as an Enhanced Enzyme Immobilization Platform for Hydrogen Peroxide Sensing” studied the synthesis of liquids-polypyrrole-gold composites and their applications of H2O2 sensing. It is systematically studied and the paper is well written. It can be accepted after the authors address the flowing comments.

Point 1: In the Synthesis of ILs-PPy composite, the addition of ammonium persulfate solution should be explained, and the reason why “placed in the dark” should also be explained.

Response 1:  The ammonium persulfate acts as initiator in the synthesis of ILs-PPy composite. The polymerization reaction must be carried out in the dark in case pyrrole auto-polymerized under light to form a large amount of oligomers. The corresponding statements have been added in the revised manuscript and were highlighted using red text (section 2.3.).

Point 2: SEM is not enough to support the large surface area of C12-PPy, BET measurement is required.

Response 2: According to your advice, we measured the BET surface area of different samples including PPy, C2-PPy, C6-PPy and C12-PPy composite, the results were provided in Figure S2 (Supporting information). The results show that the specific surface area of ILs-PPy composite increases with the alkyl chain length of ILs and C12-PPy shows largest surface area compared with others. The corresponding statements have been provided in the revised manuscript (the fourth paragraph in the section 3.1. in red text)

Point 3: What is the actual amount of Au on the C12-PPy/GCE? Is it can be controlled? Similarly, what is the actual amount of HRP on the C12-PPy/GCE?

Response 3: (1) In this paper, the amount of Au electrodeposited on the C12-PPy/GCE represents the distribution density of Au particles on the surface of C12-PPy/GCE, which can be controlled qualitatively by changing the scanning cycles of cyclic voltammetry in electrodeposition of Au on C12-PPy/GCE when the concentration of HAuCl4 solution (1 mM) and scan rate (50 mVs-1) keep constant. Generally, the distribution density of Au particles increases with the increase of scanning cycles. The obtained 30 segments (15 scanning cycles) corresponds to a larger current response, further increase the scanning cycles (more than 40 segments), the current response tends to decrease. The DPV current response of C12-PPy-Au/GCE at different scanning cycles was provided in Figure S4 (Supporting information), and the relevant explanation was provided in the revised manuscript (section 3.2. in red text).

(2) The amount of HRP is the number of HRP molecules immobilized on the surface of C12-PPy-Au/GCE. The number of HRP molecules can be controlled qualitatively by changing the concentration of HRP solution dropped on the surface of C12-PPy-Au/GCE when keep the incubating time (15 h) and incubating temperature (4oC) constant. Reasonably, the number of HRP molecules immobilized on the electrode increase with the increase of HRP concentration until reaching saturation of immobilization.

Point 4: In Figure 5A and B, it can be seen that a lot of aggregated particles on the surface. Are they Au? If they are Au, the claim of AuNPs is not suitable, as the particles are in micro-size.

Response 4: Thanks for your careful work. The aggregated particles that can be observed in Figure 5A and B is Au and thus the claim of AuNPs is not suitable indeed. The “AuNPs” and the “C12-PPy-Au nanocomposite” have been described as “Au microparticles” and “C12-PPy-Au composite” respectively in the revised manuscript.

Point 4: In Figure 9C, the concentration of HRP is 1, 2, 3, 4, 8, and 9. Why there is a big gap between 4 and 8? The optimal concentration of HRP solution of 8 may be incorrect.  In my opinion, authors should add more concentrations of HRP between 4 and 8 to get an optimal.

Response 5: Just as the reviewer mentioned that more concentrations of HRP between 4 and 8 mg·mL-1 should be added to get an optimal concentration of HRP. In fact, the HRP solution with different concentrations (including1.0, 2.0, 3.0, 4.0, 5.0, 6.0, 7.0, 8.0 and 9.0 mg·mL-1) have been examined during the optimization of HRP concentration. The results indicated that the current response of C12-PPy-Au/GCE increase significantly when the HRP concentration increase from 1.0 to 4.0 mg·mL-1. Further increase the HRP concentration, the current response of C12-PPy-Au/GCE increase slightly until the HRP concentration reach to 8.0 mg·mL-1. However, the current response of C12-PPy-Au/GCE almost keeps constant once the HRP concentration over 8 mg·mL-1. And thus the 8.0 mg·mL-1 is the optimal concentration in the present study. It was worth mentioning that the difference of current response is slightly when the HRP concentration is in the range of 4.0-8.0 mg·mL-1, and thus the corresponding data for 5.0, 6.0 and 7.0 mg·mL-1 HRP solutions have not been provided in Figure 9C. The related statements have been provided in the revised manuscript.

Point 4: There are some typos in the manuscript. eg. line 164, “absorb” should be absorbed; line 238, “know” should be “known”.

Response 4: The manuscript has been checked carefully and the appropriate corrections have been made in the revised manuscript according to the reviewer’s suggestion.

Reviewer 2 Report

Article sensors-423208 entitled “Ionic Liquids-Polypyrrole-Gold Nanocomposite as an Enhanced Enzyme Immobilization Platform for Hydrogen Peroxide Sensing” by Meng Li, Jing Wu, Haiping Su, Yan Tu, Yazhuo Shang, Yifan He, and Honglai Liu describes fabrication of GCE/PPy/Au/HPR biosensor for amperometric hydrogen peroxide determination.

Comments

1) Authors claim that after PPy synthesis in presence of [C12mim]Br “a large number of dozens-nanometer C12-PPy spheres” were obtained. However, SEM images presented in Figure 2D indicate that continuous, rough film was deposited. Therefore, in all sections of the manuscript it should be described as C12-PPy film not a nanocoposite.

2) Authors claim that in Au deposition step Au NPs were deposited. However, SEM images shown on Figure 5 indicate that Au formed aggregated structures in micrometer size. Therefore, they cannot be described as a nanoparticles but as microparticles.

3) These two above facts do not change the fact that C12-PPy-Au composite was superior in comparison to other tested materials. It enabled successful fabrication of H2O2 biosensor.

4) English grammar should be corrected by native speaker.

I recommend to accept this manuscript for publication in Sensors journal after fulfilling above comments.

Author Response

Response to Reviewer 2 Comments

Comments and Suggestions for Authors

Article sensors-423208 entitled “Ionic Liquids-Polypyrrole-Gold Nanocomposite as an Enhanced Enzyme Immobilization Platform for Hydrogen Peroxide Sensing” by Meng Li, Jing Wu, Haiping Su, Yan Tu, Yazhuo Shang, Yifan He, and Honglai Liu describes fabrication of GCE/PPy/Au/HPR biosensor for amperometric hydrogen peroxide determination. I recommend to accept this manuscript for publication in Sensors journal after fulfilling above comments.

Point 1: Authors claim that after PPy synthesis in presence of [C12mim]Br “a large number of dozens-nanometer C12-PPy spheres” were obtained. However, SEM images presented in Figure 2D indicate that continuous, rough film was deposited. Therefore, in all sections of the manuscript it should be described as C12-PPy film not a nanocoposite.

Response 1: We are appreciate for reviewer’s suggestion. “C12-PPy film” is more appropriate to describe the species formed on GCE according to SEM images, and the appropriate corrections have been made in the revised manuscript. However, the ILs-PPy samples characterized by IR and BET ect. are solid powders and thus are described as “C12-PPy composite” in the revised manuscript.  

Point 2: Authors claim that in Au deposition step AuNPs were deposited. However, SEM images shown on Figure 5 indicate that Au formed aggregated structures in micrometer size. Therefore, they cannot be described as a nanoparticles but as microparticles.

Response 2: Just as the reviewer’s mentioned that the Au particles that can be observed in Figure 5 is Au microparticles indeed. Thus the “AuNPs” and the “C12-PPy-Au nanocomposite” have been described as “Au microparticles” and “C12-PPy-Au composite” respectively in the revised manuscript.

Point 3: These two above facts do not change the fact that C12-PPy-Au composite was superior in comparison to other tested materials. It enabled successful fabrication of H2O2 biosensor.

Response 3: Thank you for your affirmation of our achievements.

Point 4: English grammar should be corrected by native speaker.

Response 4: The manuscript has been checked carefully and the appropriate corrections have been made in the revised manuscript according to the reviewer’s suggestion.

Round  2

Reviewer 1 Report

My comments are well addressed and this manuscript is OK for publication.